

# Modelled estimates of spatial variability of iron stress in the Atlantic sector of the Southern Ocean

Thomas J. Ryan-Keogh[1,2], Sandy J. Thomalla[1], Thato N. Mtshali[1], Hazel Little[2]

[1]Southern Ocean Carbon and Climate Observatory, Natural Resources and Environment, CSIR, Rosebank, Cape Town 7700, South Africa

[2]Department of Oceanography, University of Cape Town, Rondebosch, Cape Town 7701, South Africa

*Correspondence to*: Thomas.Ryan-Keogh@uct.ac.za

**Abstract**

The Atlantic sector of the Southern Ocean is characterized by markedly different frontal zones with specific seasonal and sub-seasonal dynamics. Demonstrated here is the effect of iron on the potential maximum productivity rates of the phytoplankton community. A series of iron addition productivity versus irradiance (PE) experiments utilising a unique experimental design that allowed for 24 hour incubations were performed within the austral summer of 2015/16. The addition of iron can result in the doubling of the photosynthetic parameters $\alpha^B$ and $P^B_{max}$, with subsequent changes in $E_k$. Mean values for each parameter under iron replete conditions were 1.46±0.55 ($\mu$g ($\mu$g Chl a)$^{-1}$ h$^{-1}$ ($\mu$M photons m$^{-2}$ s$^{-1}$)$^{-1}$), 72.55±27.97 ($\mu$g ($\mu$g Chl a)$^{-1}$ h$^{-1}$) and 50.84±11.89 ($\mu$M photons m$^{-2}$ s$^{-1}$); whereas mean values under the control conditions were 1.25±0.92 ($\mu$g ($\mu$g Chl a)$^{-1}$ h$^{-1}$ ($\mu$M photons m$^{-2}$ s$^{-1}$)$^{-1}$), 62.44±36.96 ($\mu$g ($\mu$g Chl a)$^{-1}$ h$^{-1}$) and 55.81±19.60 ($\mu$M photons m$^{-2}$ s$^{-1}$). There were no clear spatial patterns in either the absolute values or the absolute differences between the treatments at the experimental locations. When these parameters are integrated into a standard depth-integrated primary production model across a latitudinal transect, the effect of iron addition shows higher levels of primary production south of 50°S, with very little difference observed in the sub-Antarctic and Polar Frontal zone. These results emphasize the need for better parameterisation of photosynthetic parameters in biogeochemical models around sensitivities in their response to iron supply. Future biogeochemical models will need to consider the combined and individual effects of iron and light to better resolve the natural background in primary production and predict its response under a changing climate.



## 1. Introduction

Phytoplankton primary production (PP) in the Southern Ocean is a key contributor to global atmospheric $CO_2$ drawdown, responsible for 30-40% of global anthropogenic carbon uptake (Khatiwala et al., 2009; Mikaloff Fletcher et al., 2006; Schlitzer, 2002). High nutrient availability fuels this phytoplankton production, but growth is ultimately constrained by the lack of availability of the micronutrient iron (Fe) (de Baar et al., 1990; Martin et al., 1990). This leads to high levels of macronutrients that remain unutilised by phytoplankton growth in what is known as a High Nutrient Low Chlorophyll (HNLC) conditions. Maximum primary productivity rates of the Southern Ocean are also limited by light availability due to low incident solar angles, persistent cloud cover and deep mixed layers that curtail production and subsequently affect the efficiency of the biological carbon pump. Under future climate change scenarios, altered upwelling and mixed layer stratification (Boyd et al., 2001; Boyd and Doney, 2002), changes in sea ice cover (Arrigo et al., 2013; Montes-Hugo et al., 2008) and food-web dynamics (Dubischar and Bathmann, 1997; Moore et al., 2013; Pakhomov and Froneman, 2004; Smetacek et al., 2004) will alter both the nutrient and light supply strongly impacting primary production rates. As such, it is important that we understand the sensitivity of phytoplankton production to light and micronutrient availability so that we may improve our predictive capability of the response of the Southern Ocean carbon pump to a changing climate.

Iron plays a critical role in modulating PP due to the high requirements of the photosynthetic apparatus, photosystems I and II (Raven, 1990; Shi et al., 2007; Strzepek and Harrison, 2004). Light availability can further increase the demand for iron, as low irradiance levels increase requirements associated with the synthesis of additional photosynthetic units to increase potential light absorption (Maldonado et al., 1999; Raven, 1990; Strzepek et al., 2012; Sunda and Huntsman, 1997). Iron is also required to activate both nitrate and nitrite reductase (de Baar et al., 2005), which facilitate the assimilation of nitrate and nitrite and their subsequent intracellular reduction to ammonium. In HNLC regions, such as the Southern Ocean, nitrate uptake rates ($\rho NO_3^-$) have also frequently been reported as becoming iron limited (Cochlan, 2008; Lucas et al., 2007; Moore et al., 2013; Price et al., 1994). However, it has also been demonstrated that iron limitation rather than inhibiting nitrate reductase activity results in a bottleneck further downstream due to a reduction in photosynthetically derived reductant (Milligan and Harrison, 2000). This would lead to an excretion of excess nitrate back into the water column that would further contribute to HNLC conditions such as those present in the Southern Ocean.

Estimating PP in the oceans towards an improved understanding of the effects of iron and light limitation requires an understanding of the relationship between photosynthesis (P) and irradiance (E) (Behrenfeld and Falkowski, 1997b; Dower and Lucas, 1993; Platt et al., 2007). PE responses are derived from an equation by Platt et al. (1980), where the responses are parameterized as a function of irradiance. The parameters derived include: $P^B_{max}$, the biomass-specific rate of photosynthesis at saturating irradiances, $\alpha^B$, the irradiance-limited biomass-specific initial slope, and $E_k$, the irradiance at which saturation is initiated. The response of these parameters can be a function of temperature (Behrenfeld and Falkowski, 1997b), but also as a change in the quantum efficiency of photosynthesis, usually as the result of changes in iron availability. In previous iron fertilization experiments a doubling of $\alpha^B$ has been reported (Hiscock et al., 2008), yet this response is not consistent across Southern Ocean waters (Feng et al., 2010; Hopkinson et al., 2007; Moore et al., 2007; Smith and Donaldson, 2015). Given their relative importance within PP models (Behrenfeld and





Falkowski, 1997a, b; Sathyendranath and Platt, 2007), a greater understanding of the drivers of the variability within these photosynthetic parameters is therefore required; particularly if we are to accurately quantify and constrain PP in the Southern Ocean to examine seasonal and interannual variability and trends.

The Atlantic sector of the Southern Ocean is composed of a series of circumpolar fronts that are characterized by large geostrophic velocities (Nowlin and Klinck, 1986; Orsi et al., 1995). The fronts constrain water masses with distinct physical and chemical properties that define different oceanographic zones. These spatial zones, whilst not only displaying zonal variability with the fronts, also display important seasonal contrasts (Thomalla et al., 2011), with differing bloom initiation dates and temporal extent of bloom duration. Whilst the bloom initiation dates can in part be explained by day length and sea ice cover as you move polewards, the differences in the extent and duration of blooms between the zones requires an alternative and more nuanced explanation. One theory that has been postulated is that the supply mechanisms of iron to the mixed layer following the spring bloom varies between zones (Thomalla et al., 2011). Weak diapycnal inputs and a heavy reliance on iron recycling was suggested by Tagliabue et al. (2014) to match approximate phytoplankton utilization within the pelagic zones. An alternative theory that postulates the importance of summer storms may also be pivotal in understanding the seasonal dynamics of phytoplankton primary productivity (Nicholson et al., 2016; Swart et al., 2015; Thomalla et al., 2015), with respect to the sustained bloom observed in the Sub Antarctic Zone (SAZ). Here, summer storms are said to periodically deepen the mixed layer to below the ferricline followed by rapid shoaling during quiescent periods that balances the supply of light and iron in the upper oceans favouring phytoplankton growth that culminates in a sustained summer bloom (Swart et al., 2015). Regardless of the mechanisms at play, an understanding on when and where iron concentrations and supply mechanisms limits potential phytoplankton growth and productivity is needed to better understand the drivers that determine the characteristics of the Southern Ocean seasonal cycle.

To this end, a research cruise was conducted in the austral summer of 2015/16 as part of the third multidisciplinary *Southern Ocean Seasonal Cycle Experiment* (SOSCEx III) which aims to identify and understand the physical and chemical controls on the seasonal cycle of the biological carbon pump. As part of this study, shipboard nutrient addition PE experiments were performed to determine the extent of iron limitation upon phytoplankton primary production.




## 2. Materials and Methods

### 2.1. Oceanographic Sampling

The samples and data presented here were obtained during the 55[th] South African National Antarctic Expedition (3[rd] December 2015 to 11[th] February 2016) on-board the S.A. Agulhas II to the Atlantic sector of the Southern Ocean as part of SOSCEx III (Swart et al., 2012). During the cruise, 6 nutrient addition PE long-term experiments were performed within the Atlantic sector of the Southern Ocean (Fig. 1) to determine the extent to which relief from iron limitation could alter the maximal primary productivity rates of the phytoplankton community. Uncontaminated whole seawater was collected from 30-50 m depth using Teflon-lined, external closure 12 L Go-Flo samplers deployed on a trace metal clean CTD rosette system.

### 2.2. PE Experimental setup

Phytoplankton productivity was measured by the incorporation of $^{13}$C stable isotopes in response to an increasing light gradient. Inside a trace metal clean laboratory container, bulk trace metal clean seawater was decanted unscreened into an acid-washed 50 L LDPE carboy (Thermo scientific) to ensure homogenization; this was then redistributed into 1.0 L polycarbonate bottles (Nalgene). Sample manipulations were conducted under a laminar flow hood. All bottles were inoculated with $^{13}$C (10 µM NaH$_2$$^{13}$CO$_3$/ 100 mL) spikes to achieve an enrichment of ~5%; 11 bottles received the addition of FeCl$_3$ (+2.0 nM, 'Fe'), whereas 11 bottles received the $^{13}$C spikes alone ('Control'). The bottles were incubated in screened (LEE Filters) LDPE boxes within light and temperature controlled incubators. Experimental temperature was set to mimic the *in situ* sample collection temperature. Irradiances were measured within the screened boxes using a handheld 4$\pi$ PAR sensor (Biospherical Instruments) and ranged from 0 – 400 µM photons m$^{-2}$ s$^{-1}$. Bottles tops were covered with parafilm and double bagged with clear polyethylene bags to minimize contamination risks during the incubation.

Experiments were incubated for 24 h, after which the samples were vacuum filtered through a pre-combusted (400°C for 24 h) GF/F filter. Samples were acid fumed with concentrated HCl for 24 h to remove inorganic carbon before being dried in an oven at 40°C for 24 h. The isotopic composition of all samples were determined by mass spectrometry on a Flash EA 1112 series elemental analyser (Thermo Finnigan). Carbon uptake rates (µM C h$^{-1}$) were calculated from the equation of Dugdale and Wilkerson (1986), utilising in situ determinations of dissolved inorganic carbon (DIC). The uptake rates normalised to the chlorophyll-a (Chl) concentration, were used to calculate the maximal light-saturated Chl specific photosynthetic fixation rates (P$^{B}_{max}$), the light limited slope ($\alpha^{B}$) and the photoacclimation parameter (E$_k$). The curves and parameters were generated using a non-linear least squares fit to the equation of Platt et al. (1980).



Table 1 Locations for PE experiments conducted during the cruise along with details for the initial chemical,
physiological and physical set up conditions.

| Experiment | 1 | 2 | 3 | 4 | 5 | 6 |
|---|---|---|---|---|---|---|
| Initiation Date | 08/12/2015 | 05/01/2016 | 07/01/2016 | 08/01/2016 | 09/01/2016 | 26/01/2016 |
| Latitude (°S) | -42.69 | -42.69 | -45.99 | -50.45 | -55.70 | -70.44 |
| Longitude (°E/W) | 08.74 | 08.74 | 05.93 | 01.04 | -00.00 | -07.82 |
| Collection Depth (m) | 30 | 35 | 35 | 35 | 50 | 35 |
| Sunrise: Sunset | 03:30 – 18:30 | 04:00 – 19:00 | 04:00 – 19:00 | 04:00 – 19:00 | 04:00 – 19:00 | 00:00 – 00:00* |
| Chl (μg L$^{-1}$) | 0.97 | 0.84 | 0.89 | 2.30 | 1.15 | 1.49 |
| Nitrate (μM) | 7.21 | 10.20 | 15.83 | 21.07 | 17.02 | 23.81 |
| Silicate (μM) | 0.86 | 0.72 | 0.09 | 3.76 | 30.83 | 48.81 |
| Phosphate (μM) | 0.88 | 0.76 | 0.95 | 1.28 | 1.11 | 0.94 |
| DFe (nM) | 0.16 | 0.17 | 0.07 | 0.03 | 0.05 | 0.10 |
| $F_v/F_m$ | 0.19 | 0.30 | 0.35 | 0.30 | 0.35 | 0.37 |
| $\sigma_{PSII}$ (nm$^{-2}$) | 14.79 | 6.45 | 5.50 | 5.59 | 5.37 | 3.89 |
| MLD (m) | 33.77 | 56.96 | 108.42 | 70.11 | 42.89 | 40.80 |
| Salinity | 33.87 | 33.70 | 33.88 | 33.80 | 36.51 | 34.10 |
| Temp. (°C) | 10.80 | 10.44 | 6.72 | 3.17 | -1.42 | -1.51 |
| Average daytime PAR (μM photons m$^{-2}$ s$^{-1}$)** | 1055.31 | 787.35 | 289.18 | 524.41 | 769.87 | 673.62 |
| Euphotic Depth (m) | 72.79 | 75.10 | 52.95 | 47.92 | 69.13 | 78.07 |

*24 hour day length
**See Sect. 2.7 for details

**2.3. Chlorophyll-a and Nutrient Analysis**



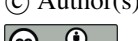

Samples for Chl analysis, 250 mL, were filtered onto GF/F filters and then extracted into 90% acetone for 24 h
in the dark at -20°C, followed by analysis with a fluorometer (TD70; Turner Designs) (Welschmeyer, 1994).
Macronutrient samples were drawn into 50 mL diluvials and stored at -20°C until analysis on land. Nitrate +
Nitrite and Silicate were measured using a Lachat Flow Injection Analyser (Egan, 2008; Wolters, 2002), whilst
Nitrite and Phosphate were determined manually by colorimetric method as specified by Grasshoff et al. (1983).
Dissolved iron samples (DFe) were carefully collected in acid-washed 125 mL LDPE bottles, acidified with
30% HCl suprapur to pH ~1.7 (using 2mL L$^{-1}$ criteria) and stored at room temperature until analysis on land at
UniBrest in France using the Chemiluminescence – Flow Injection Analyser (CL-FIA) method (Obata et al.,
1993). Accuracy and precision of the method was verified by analysis of in-house internal standards and SAFe
reference seawater samples (Johnson et al., 2007); the limits of detection were in order of 10 pM.

**2.4. Phytoplankton Photosynthetic Physiology**

Variable Chl fluorescence was measured using a Chelsea Scientific Instruments FastOcean fast repetition rate
fluorometer (FRRf) integrated with a FastAct laboratory system. Samples were acclimated in dark bottles at *in*
*situ* temperatures, and FRRf measurements were blank corrected using carefully prepared 0.2 µm filtrates for all
samples (Cullen and Davis, 2003). Protocols for FRRf measurements consisted of the following: 100 x 2 µs
saturation flashlets with a 2 µs interval, followed by 25 x 1 µs relaxation flashlets with an interval of 84 µs with
a sequence interval of 100 ms. Sequences were repeated 32 times resulting in an acquisition length of 3.2 s. The
power of the excitation LED ($\lambda450$), was adjusted between samples to saturate the observed fluorescence
transients within a given range of $R\sigma_{PSII}$. $R\sigma_{PSII}$, the probability of a reaction centre being closed during the first
flashlet, is optimised between 0.042 to 0.064 per the manufacturer specifications. By adopting this approach, it
ensures the best signal-to-noise ratio in the recovered parameters whilst accommodating significant variations in
the photophysiology of the phytoplankton community without having to adjust the protocol. Data from the
FRRf were analysed to derive fluorescence parameters as defined in Baker et al. (2001) and Roháček (2002) by
fitting transients to the model of Kolber et al. (1998).

**2.5. Pigment Analysis and CHEMTAX**

Pigment samples were collected by filtering 0.5 – 2.0 L of water onto GF/F filters. Filters were frozen and
stored at -80°C until analysis in Villefranche, France on a HPLC Agilent Technologies 1200. Filters were
extracted in 100% methanol, disrupted by sonication, clarified by filtration and analysed by HPLC following the
methods of Ras et al. (2008). Limits of detection were on the order of 0.1 ng L$^{-1}$. Pigment composition data were
standardized through root square transformation before cluster analysis utilizing multi-dimensional scaling
where similar samples appear together; and dissimilar samples do not. Samples were grouped and analysed in
CHEMTAX (Mackey et al., 1996) using the pigment ratios from Gibberd et al. (2013). Multiple iterations of
pigment ratios were used to reduce uncertainty in the taxonomic abundance as described in Gibberd et al.
(2013), with the solution that had the smallest residual used for the estimated taxonomic abundance.

**2.6. Particle Size Analysis**

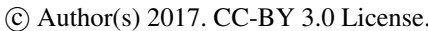




The size distribution of the particle population was measured by running 40 mL of water sample through a 100
µm aperture on a Beckman Coulter-Multisizer (20 runs at 2.0 mL per run), binning the size counts into 400 bins
between 2 µm and 60 µm. Data were subsequently analysed utilising custom Matlab scripts to calculate the
effective diameter of particles within the sample following Hansen and Travis (1974).

**2.7. Depth-integrated Production**

Water column primary production rates were calculated according to Platt et al. (1980) and Platt and
Sathyendranath (1993) as in Thomalla et al. (2015) where;

$$PP_0 = P_{max} \times (1 - e^{(\frac{-\alpha \times E_0^m \times 0.5}{P_{max}})})$$    (1)

$PP_0$ (mg C m$^{-2}$ d$^{-1}$) is the primary production at the surface, $P_{max}$ the maximal light-saturated photosynthetic
fixation rate, α the light-limited slope and $E_0^m$ is daily PAR at the surface, calculated by assuming maximum
PAR at midday, zero PAR at sunrise and sunset, a constant gradient of light between time steps and
extrapolating the measured PAR (from an above water Biospherical $4\pi$ PAR sensor) at the time of the station
into an isosceles triangle (see also Thomalla et al. (2015)).

$$E_*^m = \frac{E_0^m}{E_k}$$    (2)

The results were generalised by calculating $E_*^m$(2), the dimensionless daily surface irradiance, while primary
productivity over the entire water column $PP_{wc}$ (mg C m$^{-2}$ d$^{-1}$) was calculated with the following equation (3).
The dimensionless function $f(E_*^m)$ for daily primary productivity was solved analytically by Platt et al. (1980).
Rates were calculated for both the iron addition and control treatments, allowing the difference between the
integrated rates to be solved.

$$PP_{wc} = PP_0 \times \frac{f(E_*^m)}{k_d}$$    (3)

$K_d$ was initially calculated as the slope of the natural log of in situ PAR with depth from CTD profiles. When in
situ PAR with depth was not available, $K_d$ was also calculated from *in situ* surface Chl concentrations with the
following equation (4) (Morel, 1988; Morel et al., 2007). Co-located calculations utilising in situ PAR versus
chlorophyll-derived $K_d$ demonstrated on average a 40% higher $K_d$ when calculated with chlorophyll.

$$K_d = 0.0166 + 0.0773 \times [Chl]^{0.6715}$$    (4)

**2.8. Ancillary physical data**





Temperature and salinity profiles were obtained from a Sea-Bird CTD mounted on the rosette system. The
mixed layer depth (MLD) was calculated following de Boyer Montégut et al. (2004), which identifies the MLD
as the depth where the temperature differs from the temperature at 10 m by more than 0.2°C ($\Delta T_{10m} = 0.2°C$).
The position of the fronts was determined using sea surface height (SSH) data from maps of absolute dynamic
topography (MADT) according to (Swart et al., 2010).



## 3. Results

### 3.1. Oceanographic Context

The experimental set-up locations covered a wide range of pelagic zones from the SAZ to the Marginal Ice Zone (MIZ), each with different physical, chemical and biological properties (see Table 1). Chl concentrations between experiment initiation locations varied between $0.84 - 2.30$ µg $L^{-1}$, peaking just south of the Polar Front at ~50°S. Initial temperatures displayed a characteristic decrease from 10.80°C at the most northerly location to -1.51°C at the MIZ, whereas there were no distinct differences in salinity ranging from 33.71 to 36.51. Macronutrient concentrations all increased polewards, with peaks of 28.15 µM, 1.34 µM and 48.81 µM for nitrate, phosphate and silicate respectively. Dissolved iron concentrations decreased polewards from a maximum of 0.17 nM in the SAZ to minimum values of 0.03 nM and 0.05 nM at 50°S and 55°S respectively, before increasing again in the MIZ to 0.10 nM.

Phytoplankton photophysiology, $F_v/F_m$, increased polewards from a minimum of 0.19 to a maximum of 0.37, whereas $\sigma_{PSII}$, the effective absorption cross-section of PSII, decreased polewards from 14.79 $nm^{-2}$ to 3.89 $nm^{-2}$. The effective diameter of the phytoplankton population, a relative measure of size, increased polewards from a minimum of 4.29±0.35 µm in the SAZ to a maximum of 8.59±0.68 µm in the MIZ. Estimated taxonomic abundance through HPLC analysis and CHEMTAX determined that the dominant groups at all stations were either Diatoms, Haptophytes or a mix of the two. Haptophytes were the dominant group (>68% of total Chl) in the SAZ during experiments 1 and 2, with Diatoms becoming dominant (>70% of total Chl) from experiment 4 onwards.

MLD's were highly variable and ranged from ~34m at experiment 1 to ~108 m at experiment 3. The MLD was typically deeper than the experimental set up depth (average difference of ~15 m) at all experiments except for experiment 5 where the collection depth was 7 m below the MLD. Experiments 1 and 2 that were set up in the same location in the SAZ but 28 days apart had markedly different set up conditions; a 41% increase in the nitrate concentration from 7.21 to 10.20 µM, a two-fold increase in $F_v/F_m$ from 0.19 to 0.35 with a concurrent 56% decrease in $\sigma_{PSII}$ from 14.79 to 6.45 $nm^{-2}$ and a deepening of the MLD from ~34 m to ~57 m.

The light environment within the water column at each location was determined by calculating the percentage light depth as a function of the vertical attenuation coefficient of irradiance ($K_d$). The percentage light depths of the experiments ranged between 3.46% to 14.78%. The 1% light depth, which typically coincides with the compensation light depth i.e. the depth where rates of production equate to rates of respiration, is consistently below the MLD, except for experiment 4 where it was 22 m above the mixed layer.

### 3.2. PE Parameters

PE curves for carbon uptake ($\rho C$) (Fig. 2, Fig. S1), summarised in Table 2, display consistent results with greater values of $\alpha^B$ and $P^B_{max}$ with the addition of iron compared to the control treatments (Fig. S2). The values derived here fall within the range previously reported for iron addition experiments in the Southern Ocean (Hiscock et al., 2008; Hopkinson et al., 2007; Moore et al., 2007; Smith and Donaldson, 2015). Maximum values of $\alpha^B$ (mg C (mg Chl a)$^{-1}$ h$^{-1}$ (µmol photons m$^{-2}$ s$^{-1}$)$^{-1}$) for $\rho C$ were 2.23 x 10$^{-3}$ from experiment 2 Fe





treatment and 2.43 x $10^{-3}$ from experiment 1 control treatment, with minimum values of 0.13 x $10^{-3}$ from
experiment 5 control treatment and 0.56 x $10^{-3}$ from experiment 6 Fe treatment. $P^B_{max}$ (mg C (mg Chl a)$^{-1}$ h$^{-1}$)
values peaked in experiment 1 Fe treatment, with a minimum value of 1.06 x $10^{-2}$ in experiment 5 control
treatment. $E_k$ (μmol photons m$^{-2}$ s$^{-1}$) peaked at 79.77, with minimum values in experiment 1 control treatment.
Despite the substantial differences in set up conditions for experiments 1 and 2 in the SAZ, occupied twice over
the space of 28 days, there were no significant differences in the responses of the PE parameters to Fe.
273         To better understand the effects of iron limitation on the PE parameters, the absolute differences (Fig.
3) of $\alpha^B$, $P^B_{max}$, and $E_k$ between the iron treatments and control treatments were calculated. $\Delta\alpha^B$ ranged from -
6.94 x $10^{-4}$ to 1.30 x $10^{-3}$, with minimum and maximum percentage differences of -40.04% and 91.12%
respectively. $\Delta P^B_{max}$ ranged between 4.98 x $10^{-2}$ and -1.02 x $10^{-2}$, with minimum and maximum percentage
differences of -12.10% and 82.52%; the greatest value for $\Delta E_k$ was -40.92 for experiment 5. Maximal values of
all differences were consistently found in experiment 5 which was set up just south of the Southern Boundary
front (Figure 3).
Table 2 Summary of PE parameters, $\alpha^B$ (mg (mg Chl a)$^{-1}$ h$^{-1}$ (μmol photons m$^{-2}$ s$^{-1}$)$^{-1}$), $P^B_{max}$ (mg (mg Chl a)$^{-1}$ h$^{-1}$
$^{-1}$) and $E_k$ (μmol photons m$^{-2}$ s$^{-1}$), for the ρC nutrient addition experiments.

|  | Experiment | 1 | 2 | 3 | 4 | 5 | 6 |
|---|---|---|---|---|---|---|---|
|  | $\alpha^B$ (Fe) (x $10^{-3}$) | 1.73 | 2.23 | 1.23 | 1.56 | 1.43 | 0.56 |
|  | $\alpha^B$ (Control) (x $10^{-3}$) | 2.43 | 2.16 | 1.19 | 1.21 | 0.13 | 0.37 |
| ρC | $P^B_{max(Fe)}$ (x $10^{-2}$) | 10.67 | 9.30 | 8.46 | 6.22 | 6.04 | 2.86 |
|  | $P^B_{max(Control)}$ (x $10^{-2}$) | 9.23 | 9.14 | 9.48 | 5.99 | 1.06 | 2.56 |
|  | $E_k$ (Fe) | 61.52 | 41.72 | 68.59 | 39.80 | 42.29 | 51.12 |
|  | $E_k$ (Control) | 38.03 | 42.40 | 79.77 | 49.46 | 83.21 | 69.37 |

Potential drivers of variability within the photosynthetic parameters were determined through a Pearson's linear
correlation coefficient matrix (Fig. 4), revealing significant negative and positive relationships with sea surface
temperature (SST), salinity, nitrate and silicate concentrations, photosynthetic physiology parameters ($F_v/F_m$ and
$\sigma_{PSII}$) as well as measures of the community structure; effective diameter and ratio of Diatoms to Haptophytes.
There were no significant relationships with either dissolved iron concentrations or chlorophyll concentrations.
Other parameters that did not show any relationships and were excluded from the matrix include MLD, the light
environment (*in situ* PAR and 1% light depth) and phosphate concentrations. $\alpha^B$ for the control treatments
displayed the greatest number of relationships with SST, nitrate concentrations, community structure variables
and $F_v/F_m$. The relative differences in all the parameters showed strong positive correlations with SST and
salinity ($p<0.05$). A principle component analysis (PCA) was carried out on the data with the variables' PCA
projection on the factor plane represented in Fig. S3. The sum of the first two PC's explained 76.74% of the





total variance. The factor plane representation splits the variables, both experimental and initial conditions, into
the four different quadrants. The grouping of the variables within each quadrant agree with the positive
correlations determined within the correlation coefficient matrix; whereas variables in opposite quadrants agree
with the negative correlations.

### 3.3. Primary Production

Depth integrated primary production ($PP_{wc}$) was calculated at each experimental location and displayed a wide
range of variability with and without iron (Fig. 5). On average $PP_{wc}$ was higher in the iron addition treatments
(Fig. 5a); with an average of $387.32 \pm 207.18$ (mg C m$^{-2}$ d$^{-1}$) for iron addition and an average of $315.37 \pm 229.37$
(mg C m$^{-2}$ d$^{-1}$) for the control. The maximum absolute differences in $PP_{wc}$ ($\Delta PP_{wc}$, Fig. 5b) of 228.82 mc C m$^{-2}$
s$^{-1}$ was found in experiments 5 at ~55°S near the Southern Boundary front, with very little difference observed
in $\Delta PP_{wc}$ at experiments 3 and 4.
The responses of Fe addition to primary production from the 6 experiments were extrapolated onto
broader spatial and temporal scales, whereby underway measurements of Chl were converted into $K_d$ using
equation 4. This, when combined with underway measurements of surface PAR allowed us to look at latitudinal
gradients in primary production (as per equations 1, 2 and 3). As the PE parameters displayed strong linear
correlations with latitude, ($\alpha$ $R^2 = 0.73$ and $0.66$, $P_{max}$ $R^2 = 0.91$ and $0.68$ for Fe and Control respectively), a
linear interpolation was applied to $P_{max}$ and $\alpha$ extrapolating the values from 6 points to a 0.1° resolution along
the cruise track. The interpolated values of $P_{max}$ and $\alpha$ were combined with underway measurements of $K_d$ and
PAR to calculate $PP_{wc}$ with and without Fe addition for the three different occupations of the same transect line
(Fig. 6a). A high degree of variability was revealed between occupations in the SAZ and polar frontal zone
(PFZ) but no clear differences between the iron and control treatments. Variability in the SAZ and PFZ appears
to be temporally driven, with higher values of $PP_{wc}$ found in the third occupation of the transect line later in the
summer season. Differences in $PP_{wc}$ between the two treatments become evident south of 50°S (Fig. 6a and 6b),
with all three iron treatment occupations being ~0.5 g C m$^{-2}$ d$^{-1}$ higher than their control treatment counterparts.
The differences between the control and Fe treatments were calculated for each transect, which when combined
allowed for the calculation of an average absolute difference in primary productivity ($\Delta PP_{wc}$, Fig. 6c). $\Delta PP_{wc}$ is
slightly negative within the SAZ and PFZ, before sharply increasing to a maximum difference of 0.85 g C m$^{-2}$ d$^{-1}$
$^{-1}$ at 58°S. $\Delta PP_{wc}$ begins to decrease with increasing latitude before reaching an average difference of 0.11 g C m$^{-2}$
$^{-2}$ d$^{-1}$ in the MIZ. Representing these differences in $PP_{wc}$ as a percentage difference (Fig. 6d) shows that within
the SAZ, PFZ and MIZ the differences are ±10-20%; whereas within the Antarctic zone (55°S–65°S) the
differences between the treatments can be as much as 80%.
Given the limitations of our data set (that requires the use of interpolated values of $P_{max}$ and $\alpha$) together
with the weight we place on the conversion of these parameters to PP (with chlorophyll and PAR), it is
important that we understand the sensitivity of the PP model to variability in the different input parameters. To
test this, we performed a series of sensitivity tests to determine which components present the greatest influence
on the final PP values. The sensitivity tests were divided into the three components of the equation; $K_d$ derived
from chlorophyll (Fig. S4), surface PAR (Fig. S5) and the photosynthetic parameters ($P_{max}$ and $\alpha$) (Fig. S6). For
consistency, the range of variation for each parameter was calculated and used as a factor to alter each





component. The mean range of variability for Kd was 84.33%, surface PAR was 68.73%, and α and $P_{max}$ were
82.85% and 83.01% respectively. If $K_d$ values are increased by 84.33% this results in a 29.61% decrease in
$\Delta PP_{wc}$, whereas a decrease of $K_d$ results in an increase in $\Delta PP_{wc}$, of 59.17%. Increasing surface PAR resulted in
an increase in $\Delta PP_{wc}$ of 3.50%; whilst decreasing PAR corresponded to a decrease of 8.06%. The largest
differences in $\Delta PP_{wc}$ were generated when $P_{max}$ was altered by 83.01%, in accordance with the range of
variability, resulting in an increase of 42.97% and a decrease of 80.92% in $\Delta PP_{wc}$ (for an increase and decrease
in $P_{max}$ respectively). The other PE parameter, α, did not result in the same level of changes in $\Delta PP_{wc}$ and only
increased by 4.01% and decreased by 12.22% for an increase and decrease in α by 82.85% respectively.





## 4. Discussion

Phytoplankton biomass in the Southern Ocean is potentially limited in their extent and magnitude predominantly by the availability of the micronutrient iron (Blain et al., 2007; Boyd et al., 2000; Pollard et al., 2009). This conclusion is based on the combination of two factors, the high iron requirements for photosynthetic proteins (Shi et al., 2007) and the lack of supply sources of iron to the Southern Ocean (Duce and Tindale, 1991; Tagliabue et al., 2014). The result of which is an environment that displays high degrees of spatial and temporal variability in primary production in response to highly variable iron supply mechanisms that result in chlorophyll patchiness (Fig. 1) and a complex seasonality (Thomalla et al., 2011). Iron limitation is potentially strongest during the summer months when light levels are not considered limiting and the spring bloom is expected to have utilised the bulk of the winter iron resupply. In the austral summer of 2015/2016 a series of iron addition photosynthesis versus irradiance experiments were performed in the Atlantic Southern Ocean to determine the extent to which iron availability was limiting maximal rates of primary productivity.

The addition of iron appeared to stimulate increased productivity to varying degrees (Fig. 2, Fig. 3b, Fig. S1, Fig. S2) with average $P_{max}$ and $\alpha$ values being higher for an iron replete system ($12.75 \pm 6.95$ and $0.25 \pm 0.14$) compared to a control system ($11.17 \pm 8.23$ and $0.22 \pm 0.19$), suggestive that iron is indeed a micronutrient limiting phytoplankton production in this region. Similar responses have been reported by Hiscock et al. (2008) under conditions of sub-saturating light conditions, where the addition of iron can result in a doubling of photosynthetic rates. However, a nutrient addition PE experiment in the Ross Sea demonstrated no significant increases in $\alpha^B$ or $P^B_{max}$ (Smith and Donaldson, 2015). One potential reason for this is the length of their incubation period, which was only 2 hours and may not have been sufficient enough for the phytoplankton to incorporate the iron into their proteins and produce higher productivity rates. Indeed, nutrient addition experiments performed under similar conditions were shown to require 24 hours to see any significant differences in photophysiology (Ryan-Keogh et al., 2017; Ryan-Keogh et al., 2013) with changes in biomass only being reported after 48 hours. This shortcoming highlights the attraction of the unique experimental design utilised here, which allows for 24-hour Fe addition and control incubations at varying light levels and constant temperature.

Potential factors that are known to be associated with iron-induced enhanced primary productivity include temperature, macronutrient concentrations, Chl, MLD, light history and community composition. A Pearson's linear correlation matrix (Fig. 4) was carried out on an array of variables to examine the influence of key physical, chemical and biological factors on the variability of photosynthetic parameters in this study. Significant relationships were found with SST, salinity and macronutrient concentrations, which show strong latitudinal gradients. A proxy for the community structure that utilized the ratio of the 2 dominant groupings (Diatoms and Haptophytes) also indicated strong significant relationships with the PE parameters. No significant relationships were however found between PE parameters and iron or Chl concentrations. The lack of significant relationships could be due to the small range of variability observed in these parameters; for example, Chl concentrations at all stations were typically low ($0.84 - 2.30$ (µg L$^{-1}$) as were dissolved iron concentrations ($0.03 - 0.17$ nM). The lack of a relationship with dissolved iron concentrations highlights how this proxy is not necessarily a good indicator of iron stress, as any limiting nutrient would be expected to be





severely depleted by biological uptake with a resultant ambient concentration that would remain close to zero
despite possible event scale supply (Ryan-Keogh et al., 2017).
The photosynthetic parameters derived here are important components in a suite of models that derive
estimates of phytoplankton primary production (Behrenfeld and Falkowski, 1997a, b; Sathyendranath and Platt,
2007). Different primary production models inherently consist of certain biases towards modelling the
photosynthetic parameters whereas others have excluded them entirely from the computation of primary
productivity rates. Hiscock et al. (2008) demonstrated that the variables in the Behrenfeld and Falkowski
(1997b) standard depth-integrated model (DIM) exerted considerably different forcing mechanisms on the final
primary productivity rates. In the case of this DIM, phytoplankton biomass was the dominant variable that could
result in three orders of magnitude changes in primary production, compared to only a 40-fold change when
altering the photosynthetic parameter $P^B_{opt}$ (i.e. $P^B_{max}$). This highlights the need to understand the sensitivity of
different PP models to variability within their input parameters.
Results from the production model applied here (equations 1, 2 and 3) show a general decrease with
latitude in depth-integrated primary production ($PP_{wc}$), with significant differences between treatments (t-test,
$p<0.05$). One station near the Southern Boundary exhibited the greatest differences in $\Delta PP_{wc}$ with a value of
0.89 g C m$^{-2}$ d$^{-1}$ (Fig. 5b), with the lowest observed $\Delta PP_{wc}$ of 0.11 g C m$^{-2}$ d$^{-1}$ south of the polar front. The low
sampling frequency of the experiments both spatially and temporally (6 experiments spanning two months and
the entire latitudinal extent of the Southern Ocean) together with the diverse range of initial set up conditions
(Table 1) make it difficult to interpret the causal relationships observed within each experiment with any
certainty. Instead, the information from these experiments were maximised through an alternate approach that
utilised the range of variability in PE parameters in control versus iron addition experiments to gain a broader
spatial interpretation of the response of phytoplankton production to iron addition.
A linear interpolation of the PE parameters ($P_{max}$ and $\alpha$) with latitude, together with underway
measurements of PAR and $K_d$ (derived from surface Chl) allow for the generation of high resolution rates of
$PP_{wc}$ with and without Fe addition for three occupations of the cruise transect (Fig. 6a). Within the SAZ and
PFZ there was a high degree of variability between the three occupations, with higher $PP_{wc}$ values later in the
growing season (Fig. 6a). However, there were no clear differences between the iron and control treatments in
any of the occupations. Whereas south of 50°S there were no differences as the growing season progressed but a
clear difference between the iron and control treatments (Fig. 6b and 6c). Here, a maximum percentage
difference of ~80% (Fig. 6d) was observed between control and iron replete conditions, with $\Delta PP_{wc}$ peaking at
0.85 g C m$^{-2}$ y$^{-1}$ at 55°S. Differences between iron addition and control systems begin to decline within the MIZ
(Fig. 6c). These results suggest that there are potential differences in iron availability and supply within different
zones of the Southern Ocean, which agrees with previous studies which postulated that the bloom extent and
duration within the SAZ could potentially be driven by enhanced iron supply through storm-eddy interaction
(Nicholson et al., 2016) while in the MIZ addition iron is supplied through melting ice (Gao et al., 2003; Grotti
et al., 2005; Sedwick and DiTullio, 1997). The Fe addition test performed here demonstrates the sensitivity of
waters south of 50°S to Fe availability, if models do not consider this sensitivity then the degree of error for PP
models can be as high as 80%.
From these results, it became clear that higher values of $P_{max}$ and $\alpha$ because of iron addition were
significantly influencing the model outputs of primary production. However, the extent to which changes in the



PE parameters were responsible for the latitudinal trend in $\Delta PP_{wc}$ versus changes in ancillary parameters (e.g.
Chl, PAR) is unclear. To test our interpretation of the variability in $PP_{wc}$ being a direct response to Fe
availability through changes in the PE parameters, a series of sensitivity analyses were performed which showed
that PAR and $\alpha$ exerted very little influence (Fig. S5 and S6). Biomass (Chl), as represented through $K_d$, did
exert a large influence on $PP_{wc}$ (up to 59%, Fig. S4), however the greatest influence was $P_{max}$ (up to 81%, Fig.
S6). As such, we can conclude that the primary driver of the latitudinal trend in $\Delta PP_{wc}$ is the result of changes in
the maximum photosynthetic capacity ($P_{max}$) to iron addition.

429         The photosynthetic parameters $P_{max}$ and $\alpha$ remain difficult to fully parameterise due to interacting

effects of iron, light availability, temperature and community structure, yet these parameters remain critical
components of different biogeochemical models. Our results show that if models fail to capture the interacting
effects of iron and other parameters on primary productivity, then the degree of error across vast extents of the
Southern Ocean can be significant (as much as 80%). On the other hand, any model that can correctly account
for variability in these parameters will better reproduce the natural background levels of primary productivity
and the seasonal cycle for application to iron limited areas of the ocean including the Sub-Arctic Pacific and the
Southern Ocean.

**Acknowledgements**

We would like to thank the South African National Antarctic Programme (SANAP) and the captain and crew of
the SA Agulhas II for their professional support throughout the cruise. Ryan Cloete and Ryan Miltz were
involved in experimental set up; Natasha van Horsten and Warren Joubert performed the DIC determinations for
calculation of carbon assimilation. This work was undertaken and supported through the CSIR's Southern
Ocean Carbon and Climate Observatory (SOCCO) Programme (http://socco.org.za/). This work was supported
by CSIR's Parliamentary Grant funding and the NRF SANAP grant (SNA14073184298).



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





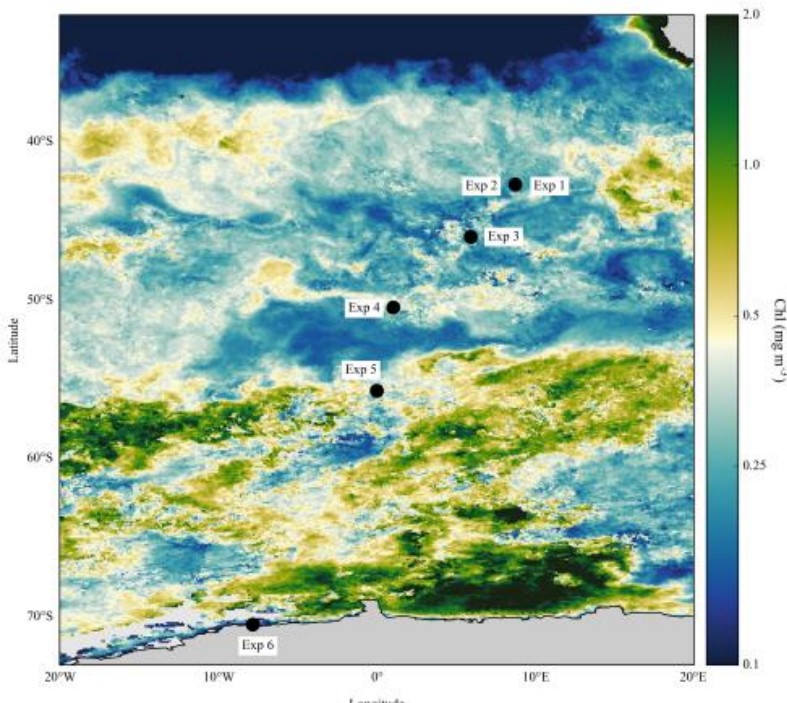


**Figure 1: Composite map of MODIS (8-day, 9 km) derived chlorophyll (mg m$^{-3}$) from November 2015 to March 2016 for the Atlantic sector of the Southern Ocean with locations of the nutrient addition productivity versus irradiance (PE) experiments.**







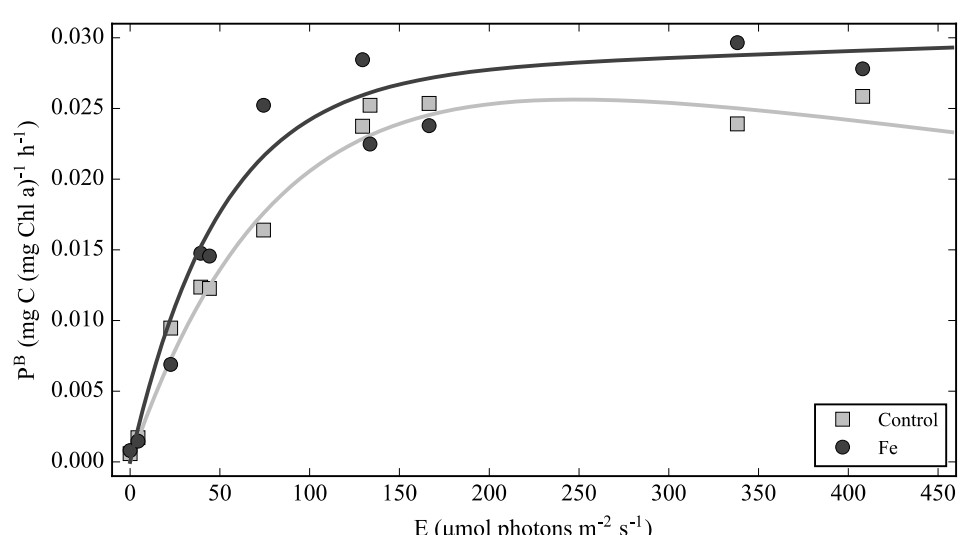


**Figure 2: An example of a PE curve of productivity (mg C (mg Chl a)$^{-1}$ h$^{-1}$), versus irradiance (μmol photons m$^{-2}$ s$^{-1}$), with (Fe) and without (Control) the addition of iron; the lines represent a non-linear least squares fit to the equation of Platt et al. (1980).**






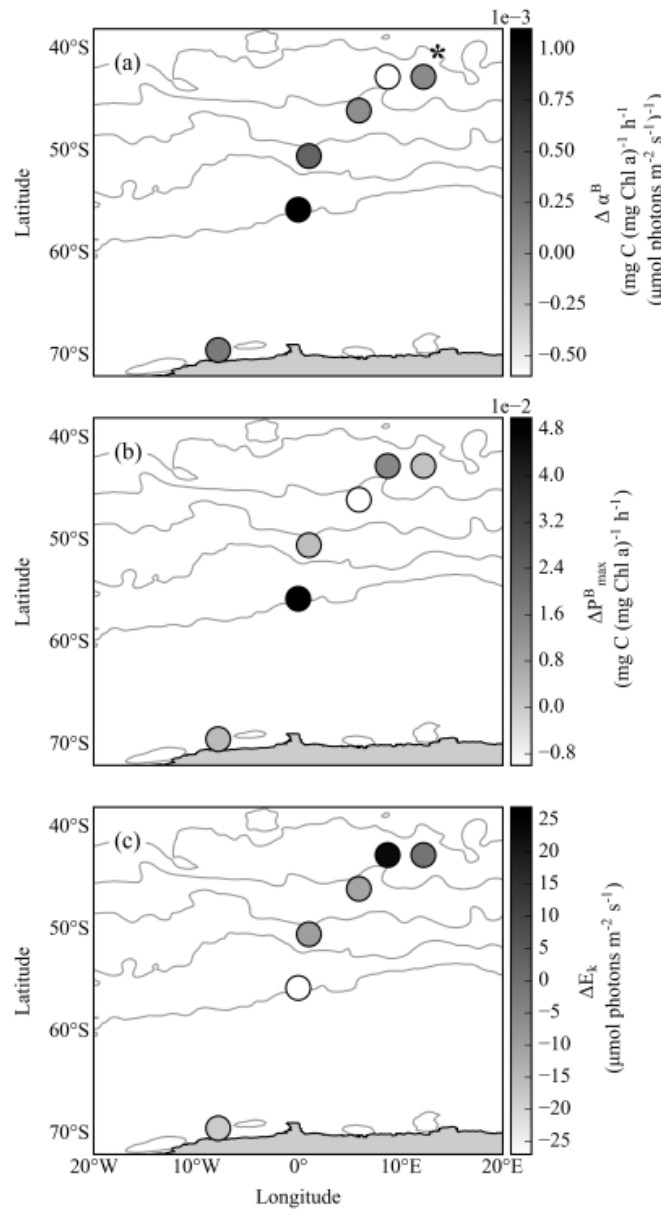


**Figure 3: Experimental values of (a) $\Delta\alpha^B$ (mg C (mg Chl a)$^{-1}$ h$^{-1}$ ($\mu$mol photons m$^{-2}$ s$^{-1}$)$^{-1}$), (b) $\Delta P^B_{max}$ (mg C (mg Chl a)$^{-1}$ h$^{-1}$) and (c) $\Delta E_k$ ($\mu$mol photons m$^{-2}$ s$^{-1}$) for experiments set up in the Atlantic sector of the Southern Ocean. Ocean fronts, indicated by grey lines, were determined from MADT from the CLS/AVISO product (Rio et al., 2011) and their position averaged over 5 months (November 2015 to March 2016). From north – south: Sub-Tropical Front (STF), Sub-Antarctic Front (SAF), Antarctic Polar Front (APF), Southern Antarctic Circumpolar Front (SACCF) and the Southern Boundary (SBdy). *Position of experiment 3 moved 2.5° eastwards for presentation purposes.**

675





676

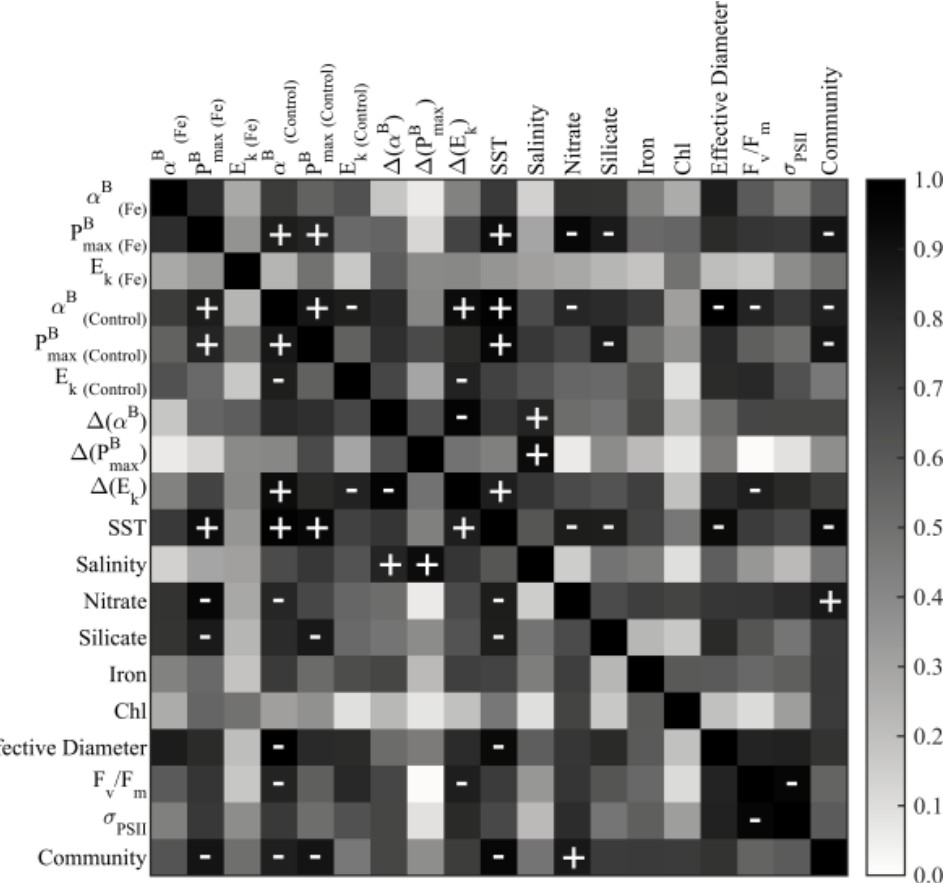

677

**Figure 4: Matrix of Pearson's linear correlation coefficients between the photosynthetic parameters determined experimentally and *in situ* variables measured, including: $\alpha^B$, $P^B_{max}$ and $E_k$ from the both Fe and control treatments, the relative differences, sea surface temperature (SST), Salinity, Nitrate, Silicate and dissolved Iron concentration, Chl concentration, Effective Diameter, $F_v/F_m$, $\sigma_{PSII}$ and Community composition (ratio of Diatoms to Haptophytes). The strength of the linear relationship associated between each pair of variables is indicated by the colour of the square, with the negative and positive correlations denoted by '-' and '+' within all squares where significant ($p<0.05$).**





Figure 5

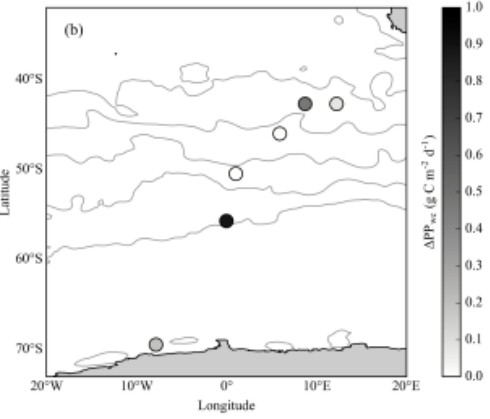


**Figure 5: Modelled outputs of primary production utilizing experimentally derived photosynthetic parameters. (a)**
**Depth integrated primary production (PP$_{wc}$) (mg C m$^{-2}$ d$^{-1}$) and (b) ΔPP$_{wc}$ (mg C m$^{-2}$ d$^{-1}$). Ocean fronts, indicated by**
**grey lines, displayed as in Fig. 3.**






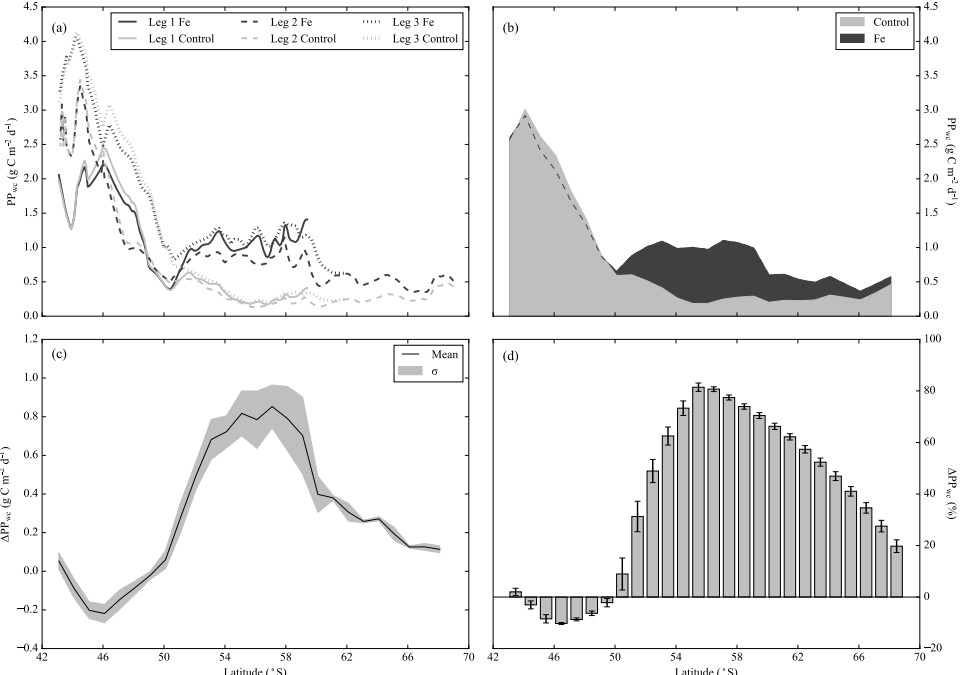


**Figure 6: Depth integrated primary production (PPwc) (mg C m⁻² d⁻¹) for each transect (Leg 1 -3) (a) interpolated along the transect line utilizing linearly interpolated values for α and Pmax as determined from the Fe and Control treatments. (b) Mean PPwc (mg C m⁻² d⁻¹) with ± standard deviation (σ). (c) The mean absolute differences in PPwc (ΔPPwc) with ± standard deviation between the Fe and Control treatments. (d) ΔPPwc represented as the mean percentage difference with ± standard deviations.**

700