# Peer review of "Modelled estimates of spatial variability of iron stress in the"

_Biogeosciences, 2017_

## Referee Comment (RC1) · D. A. Hutchins (Referee) · 26 Apr 2017

General comments

This study examines the responses of phytoplankton photosynthesis versus irradiance parameters to iron additions across a Southern Ocean transect, and addresses an important set of classic questions about iron and light limitation interactions in this region. Although quite a number of related studies have been performed throughout the region in the past, this one is unusual in incorporating such a long latitudinal transect spanning the major Southern Ocean biogeochemical provinces. The authors also attempt to integrate their results with a simple model describing the influence of iron and light on primary productivity. In general, the work was carefully done and the results are

definitely worthy of publication.

Their iron addition incubations during the PE curve experiments were deliberately short, only 24 hours, in order to avoid changes in biomass. While this is probably long enough to see initial changes in photophysiology, as they note on p. 13, it does seem possible that iron-mediated photosynthetic responses might not be fully completed in that time. More importantly, the increases in biomass and shifts in communities that would have happened had they drawn their iron addition experiments out longer are important to consider too- as they say when discussing the results of their sensitivity analysis on lines 425-426, 'Biomass... did exert a large influence on PP (up to 59%...)'. Since their incubations weren't long enough to evaluate this biomass increase, the +Fe depth-integrated productivity calculations and model they present in Figs. 5 and 6 are undoubtedly considerably lower than would be the case if the community experienced sustained relief from Fe limitation. I think the results and conclusions obtained here need to be qualified as applying only to the initial responses of these communities to iron additions, and it should be explicitly recognized in the text that they cannot be applied to understand longer term community responses (which would also include taxonomic composition shifts).

Likewise, the fact that in some of these short iron addition experiments PE parameters did not change (for instance, stations 1 and 2 in the SAZ, p. 10 lines 271-272 and Table 2) cannot be taken as evidence for lack of iron limitation at these stations. Much longer duration SAZ iron addition experiments published way back in 2001 show similar lack of changes in alphaB and PBmax, but in those same experiments the iron additions led to much higher biomass and to community composition changes- clear evidence that the community actually was iron-limited (Hutchins et al. 2001 JGR 106). Many other past studies have also conclusively confirmed ecological and physiological iron limitation of SAZ communities; one of the most highly cited is the SoFex in situ iron fertilization study of Coale et al. (2004, Science 304).

By the way, these older papers also carefully considered the effects of Si limitation

(or iron/Si co-limitation) of diatoms in the SAZ, which is not considered at all in the present paper. Clearly though, the gradient in Si availability this study covered was much greater than any gradient in N, P, or even Fe (Table 1, p. 5), and this is probably the reason that according to their pigment analyses haptophytes were dominant in the SAZ and diatoms south of the Polar Front (lines 245-247). Could changes in the makeup of these communities driven by Si availability have any influence on their PE results? This is probably worth considering briefly in the discussion. Finally, as they briefly acknowledge on line 398, this cruise spanned two full months, and so in practice examined a seasonal shift as well as a latitudinal gradient. The relative importance of iron and light limitation (yes and Si limitation!) changes across the growing season in different ways in the various Southern Ocean regimes they examined (see a simplified diagram of this seasonal pattern in Fig 2 of Boyd et al 2010, L&O 55). It would be worthwhile to discuss this aspect of their study in more detail in the text.

Specific comments

Abstract lines 19-22. These quantitative values need to be better linked to the specific photosynthetic parameter each belongs to, it requires quite a bit of peering back and forth for the reader to figure out which number goes with which parameter. A simple rewording would be helpful.

Line 43. The Arrigo et al. 2013 reference on ice cover changes given here deals with the Arctic, not the Antarctic, and should be replaced.

Line 123 and Figure 2. Obviously the maximum irradiance of 400 used in the PE curves was still below photoinhibiting levels. It would have been interesting (if logistically challenging, as I admit!) to extend it out to higher irradiances to add some perspective on this end of the curve.

Line 171, section 2.5. Another thing I wish the authors had done was to get better taxonomic information than can be obtained by the simple class-level distinctions possible through HPLC pigment measurements along with CHEMTAX. Just because there

were diatoms all along their transect doesn't mean they were ecologically or biogeo-chemically equivalent. In fact, SAZ diatoms tend to be small, delicate, lightly silicified pennates while diatoms south of the Polar Front are typically much more robust and silicified, and much more likely to be significant in export. It seems a shame to do all this work, and then be limited in the wider inferences that can be drawn due to having only bulk measurements of productivity and broad general classes of phytoplankton. Some more detailed taxonomic and functional information would have made the paper more useful and interesting.

Line 348. The Shi et al. 2007 reference is a study on the tropical N2-fixing cyanobacterium Trichodesmium, and is not appropriate here. Please add a study on Southern Ocean phytoplankton, or at least on eukaryotic phytoplankton in general.

---

## Referee Comment (RC2) · B. Quéguiner (Referee) · 28 Apr 2017

This paper addresses the long-standing question of the role of iron availability on the photosynthetic parameters of naturally occurring populations of plankton in the Atlantic sector of the Southern Ocean. Only 6 stations are studied, approximately positioned in the SAZ, the PFZ, the Antarctic zone and the MIZ. The experiments consisted of producing photosynthesis-light curves over 24 hours of incubation under iron enrichment conditions vs. no enrichment. There are (too) many major methodological problems associated with these results.

1) No details are given on the pre-treatment of the incubation vials (ultra-clean conditions?) 2) The duration of the incubations (24 hours) does not make it possible to

obtain an estimate of the in situ photosynthetic parameters of the natural phytoplankton because it is known that the adaptation time of these parameters in response to a change in light regime is on the order of the 2 to 6 hours. Within 24 hours, each incubated sample thus has ample time to adapt to the light intensity at which it is incubated. Nevertheless, these experimental values are used by the authors (apparently unaware of this major problem of different time scales between light acclimation and iron relief) in an extrapolation across the entire Atlantic area in order to evaluate the primary production of this sector. 3) The sampling strategy is curious with one of the samples (station 5) collected under the mixed layer. In addition, at the end of the manuscript, one discover that there were three occupations of the transect with a total 6 stations sampled (?) 4) There is no measurement of DFe in the samples themselves. What about contaminations of controls? 5) The measurements are not carried out in triplicate and it is therefore impossible to evaluate the precision in the estimation of the photosynthetic parameters. 6) Inconsistencies in the determination of the light attenuation coefficient (40% difference between the PAR profile and the empirical equation as a function of chlorophyll). It is then not known which estimate is used in the primary production model.

Also many problems in the expression of results: 1) a salinity change of 33.71 to 36.51 is considered "not distinct" (line 235). A chlorophyll range between 0.84 and 2.3 is considered as a "small range of variability", with individual values considered to be "low", indicating a total ignorance of the oceanography of this region. 2) The presentation of the photosynthetic parameters (paragraph 3.2.) is surprising. It is written: "PE curves for carbon uptake (C) (Fig. 2, Fig. S1), summarised in Table 2, display consistent results with greater values of $\alpha$B and PB with the addition of iron compared to the control treatments (Fig. S2).", which is completely inconsistent with Figure S2. Moreover, the choice of the figure showing the effect of iron on the P-E curve (figure 2) is at the limit of what is acceptable: it is the only good relation of this type on the 6 curves presented on figure S2. On top of that I do not understand how the parameters have been inferred from experimental data. I defy anybody to see significant changes in the photosynthetic

parameters, related to iron enrichment, from the curves presented as supplementary material.

---

## Author Comment (AC1) · 1 Jun 2017

General comments This study examines the responses of phytoplankton photosynthesis versus irradiance parameters to iron additions across a Southern Ocean transect, and addresses an important set of classic questions about iron and light limitation interactions in this region. Although quite a number of related studies have been performed throughout the region in the past, this one is unusual in incorporating such a long latitudinal transect spanning the major Southern Ocean biogeochemical provinces. The authors also attempt to integrate their results with a simple model describing the influence of iron and light on primary productivity. In general, the work was carefully

done and the results are definitely worthy of publication. Their iron addition incubations during the PE curve experiments were deliberately short, only 24 hours, in order to avoid changes in biomass. While this is probably long enough to see initial changes in photophysiology, as they note on p. 13, it does seem possible that iron-mediated photosynthetic responses might not be fully completed in that time. More importantly, the increases in biomass and shifts in communities that would have happened had they drawn their iron addition experiments out longer are important to consider too- as they say when discussing the results of their sensitivity analysis on lines 425-426, 'Biomass: : : did exert a large influence on PP (up to 59%...)'. Since their incubations weren't long enough to evaluate this biomass increase, the +Fe depth-integrated productivity calculations and model they present in Figs. 5 and 6 are undoubtedly considerably lower than would be the case if the community experienced sustained relief from Fe limitation. I think the results and conclusions obtained here need to be qualified as applying only to the initial responses of these communities to iron additions, and it should be explicitly recognized in the text that they cannot be applied to understand longer term community responses (which would also include taxonomic composition shifts).

A statement to this effect has been added to the discussion to highlight that these results only reflect initial responses and do not take into account community shifts and longer term responses. Please see p. 13 lines 371-373. "However, it should be noted that a time-length of 24 hours may not be sufficient to complete alleviate the iron-mediated photosynthetic response and as such these results may only reflect initial responses rather than longer term community level responses to relief from iron limitation."

Likewise, the fact that in some of these short iron addition experiments PE parameters did not change (for instance, stations 1 and 2 in the SAZ, p. 10 lines 271-272 and Table 2) cannot be taken as evidence for lack of iron limitation at these stations. Much longer duration SAZ iron addition experiments published way back in 2001 show similar lack of changes in alphaB and PBmax, but in those same experiments the iron additions

led to much higher biomass and to community composition changes- clear evidence that the community actually was iron-limited (Hutchins et al. 2001 JGR 106). Many other past studies have also conclusively confirmed ecological and physiological iron limitation of SAZ communities; one of the most highly cited is the SoFex in situ iron fertilization study of Coale et al. (2004, Science 304).

A statement has been added to discuss the potential for longer term relief from iron addition that may not have been achieved in this study. Please see p. 14 lines 410-412. "This may not reflect a lack of iron limitation in the SAZ, as it has been demonstrated previously that there is ecological and physiological iron limitation (Coale et al., 2004), with longer experiments demonstrating increases in Pmax and $\alpha$ following iron addition (Hutchins et al., 2001)."

By the way, these older papers also carefully considered the effects of Si limitation (or iron/Si co-limitation) of diatoms in the SAZ, which is not considered at all in the present paper. Clearly though, the gradient in Si availability this study covered was much greater than any gradient in N, P, or even Fe (Table 1, p. 5), and this is probably the reason that according to their pigment analyses haptophytes were dominant in the SAZ and diatoms south of the Polar Front (lines 245-247). Could changes in the makeup of these communities driven by Si availability have any influence on their PE results? This is probably worth considering briefly in the discussion.

A statement to address this has been added to the discussion, see p. 13 lines 389-397. "A proxy for the community structure that utilized the ratio of the 2 dominant groupings (Diatoms and Haptophytes) also indicated strong significant relationships with the PE parameters, which is potentially driven by Si availability controlling community structure. Indeed, it has been demonstrated that in the SAZ, where haptophytes dominated during this study, there is evidence for Fe-Si co-limitation. In a study by Hutchins et al. (2001) it was demonstrated that the addition of both Fe and Si resulted in the greatest responses in chlorophyll and the photosynthetic parameters. The relationship here may not be driven by Fe availability on the PE parameters, but rather community level

limitation."

Finally, as they briefly acknowledge on line 398, this cruise spanned two full months, and so in practice examined a seasonal shift as well as a latitudinal gradient. The relative importance of iron and light limitation (yes and Si limitation!) changes across the growing season in different ways in the various Southern Ocean regimes they examined (see a simplified diagram of this seasonal pattern in Fig 2 of Boyd et al 2010, L&O 55). It would be worthwhile to discuss this aspect of their study in more detail in the text.

The seasonal shifts in limitations have been added to discussion, please see p. 15 lines 477-482 and lines 494-495. "It must be noted that the transects will not only reflect latitudinal gradients but will also contain a seasonal signal as the cruise spanned 2 months across the austral summer. A seasonal shift in community structure of haptophytes increasing their dominance beyond the SAZ into the PFZ was evident from underway measurements of community structure (data not shown); indicative of seasonal Si limitation for this region (Boyd et al., 2010). Moreover, the complex seasonality of this region represents shifts between varying co-limitations that will be represented not only in the PE parameters measured but also in the additional components utilized to calculate PPwc."

"As such, we can conclude that the primary driver of the latitudinal trend in $\Delta$PPwc is the result of changes in the maximum photosynthetic capacity (Pmax) to iron addition, however, regions along the transect may be experiencing seasonal co-limitation of Fe and Si, particularly during the third transect conducted during late summer."

Specific comments

Abstract lines 19-22. These quantitative values need to be better linked to the specific photosynthetic parameter each belongs to, it requires quite a bit of peering back and forth for the reader to figure out which number goes with which parameter. A simple rewording would be helpful.

**BGD**

Wording has been restructured to make it clearer, see p. 1 lines 18-22. "A series of iron addition productivity versus irradiance (PE) experiments utilising a unique experimental design that allowed for 24 hour incubations were performed within the austral summer of 2015/16 to determine the photosynthetic parameters $\alpha$B, PBmax and Ek. Mean values for each photosynthetic parameter under iron-replete conditions were $\alpha$B: 1.46$\pm$0.55 ($\mu$g ($\mu$g Chl a)-1 h-1 ($\mu$M photons m-2 s-1)-1), PBmax: 72.55$\pm$27.97 ($\mu$g ($\mu$g Chl a)-1 h-1) and Ek: 50.84$\pm$11.89 ($\mu$M photons m-2 s-1); whereas mean values under the control conditions were $\alpha$B: 1.25$\pm$0.92 ($\mu$g ($\mu$g Chl a)-1 h-1 ($\mu$M photons m-2 s-1)-1), PBmax: 62.44$\pm$36.96 ($\mu$g ($\mu$g Chl a)-1 h-1) and Ek: 55.81$\pm$19.60 ($\mu$M photons m-2 s-1)."

Line 43. The Arrigo et al. 2013 reference on ice cover changes given here deals with the Arctic, not the Antarctic, and should be replaced.

The Arrigo et al. 2013 reference has been removed and the following references have been inserted into this section.

Close, S. E. and Goosse, H.: Entrainment-driven modulation of Southern Ocean mixed layer properties and sea ice variability in CMIP5 models, Journal of Geophysical Research-Oceans, 118, 2811-2827, 10.1002/jgrc.20226, 2013.

de Lavergne, C., Palter, J. B., Galbraith, E. D., Bernardello, R., and Marinov, I.: Cessation of deep convection in the open Southern Ocean under anthropogenic climate change, Nature Climate Change, 4, 278-282, 10.1038/nclimate2132, 2014.

Zhang, J. L.: Increasing Antarctic sea ice under warming atmospheric and oceanic conditions, Journal of Climate, 20, 2515-2529, 10.1175/jcli4136.1, 2007.

Line 123 and Figure 2. Obviously the maximum irradiance of 400 used in the PE curves was still below photoinhibiting levels. It would have been interesting (if logistically challenging, as I admit!) to extend it out to higher irradiances to add some perspective on this end of the curve.

[Figure]

Indeed, the limitations of the incubator set up did prevent us from being able to determine potential levels of photoinhibition. Future experiments planned will encompass a change in experimental set up to try and achieve higher irradiances. Please see p. 10 lines 274-276. "Due to constraints in light levels for the incubator set up, light levels that may result in photoinhibition (>400 $\mu$mol photons m-2 s-1) were not achieved and as such no measurements of $\beta$ were determined."

Line 171, section 2.5. Another thing I wish the authors had done was to get better taxonomic information than can be obtained by the simple class-level distinctions possible through HPLC pigment measurements along with CHEMTAX. Just because there were diatoms all along their transect doesn't mean they were ecologically or biogeochemically equivalent. In fact, SAZ diatoms tend to be small, delicate, lightly silicified pennates while diatoms south of the Polar Front are typically much more robust and silicified, and much more likely to be significant in export. It seems a shame to do all this work, and then be limited in the wider inferences that can be drawn due to having only bulk measurements of productivity and broad general classes of phytoplankton. Some more detailed taxonomic and functional information would have made the paper more useful and interesting.

We agree with the reviewer that further taxonomic data would enhance certain aspects of this paper, as such a further study is being conducted using microscopy counts alongside coulter counter and HPLC data. However, this data analysis is ongoing and will not be available for this manuscript.

The taxonomic data we do report however states that the SAZ is dominated by haptophytes not diatoms. In addition we also present information on the dominant size structure from coulter counter effective diameter, which ranges latitudinaly from a minimum in the SAZ (4.29 um) to a maximum in the MIZ (8.59 um)– please see p. 9 lines 249 – 256. Diatoms only become dominant from experiment 4 onwards with subsequent changes in effective diameter.

Line 348. The Shi et al. 2007 reference is a study on the tropical N2-fixing cyanobacterium Trichodesmium, and is not appropriate here. Please add a study on Southern Ocean phytoplankton, or at least on eukaryotic phytoplankton in general.

Changed the reference to Raven 1990, Twining & Baines 2013, Quigg et al. 2003, Strezpek and Harrison 2004.

---

## Author Comment (AC2) · 1 Jun 2017

This paper addresses the long-standing question of the role of iron availability on the photosynthetic parameters of naturally occurring populations of plankton in the Atlantic sector of the Southern Ocean. Only 6 stations are studied, approximately positioned in the SAZ, the PFZ, the Antarctic zone and the MIZ. The experiments consisted of producing photosynthesis-light curves over 24 hours of incubation under iron enrichment conditions vs. no enrichment. There are (too) many major methodological problems associated with these results.

1) No details are given on the pre-treatment of the incubation vials (ultra-clean conditions?)

All experimental conditions were carried out in a class-100 clean container. Experimental bottles were pre-treated with detergent and acid (Hydrochloric) as per trace metal clean standards. Please see p.4 lines 115 – 118.

"Inside a trace metal clean laboratory class-100 container, bulk trace metal clean seawater was decanted unscreened into an acid-washed 50 L LDPE carboy (Thermo scientific) to ensure homogenization; this was then redistributed into acid-cleaned 1.0 L polycarbonate bottles (Nalgene). All experimental conditions were conducted and carried out following trace metal clean standards and conditions."

2) The duration of the incubations (24 hours) does not make it possible to obtain an estimate of the in situ photosynthetic parameters of the natural phytoplankton because it is known that the adaptation time of these parameters in response to a change in light regime is on the order of the 2 to 6 hours. Within 24 hours, each incubated sample thus has ample time to adapt to the light intensity at which it is incubated. Nevertheless, these experimental values are used by the authors (apparently unaware of this major problem of different time scales between light acclimation and iron relief) in an extrapolation across the entire Atlantic area in order to evaluate the primary production of this sector.

The reviewer here is highlighting that the incubation time would make these measurements unsuitable estimates for the levels of community PP in the Southern Ocean, which we agree with. The results are designed to highlight the potential differences between the treatments, rather than absolute numbers of PP. As any long-term changes in iron limitation could ultimately lead to community structure change or potential secondary limitation by silica for example.

Light acclimation can change the iron quota of in situ phytoplankton – higher light is expected to decrease the iron quota in Antarctic phytoplankton species (Strzepek et al., 2012). However, the light ranges of the experiments (0 – 400) fall below the maximum

light intensities measured in situ at the time of the experimental set up (see Table 1). This would suggest that the design would be expected to increase the iron demand and we would expect to see larger increases in PP following iron addition.

Statements to highlight this shortcoming have been added into the text, see p. 13 lines 420-423.

"It should be noted however, that light acclimation can occur on time scales of between 2 – 6 hours and as such be reflected in the potential iron demand, with a lower demand expected at higher irradiances (Strzepek et al., 2012). Such incidences would impact the observed differences between PE parameters in control versus Fe addition experiments. However, since the light ranges of the experiments (0 – 400) fall below the maximum light intensities measured in situ (Table 1), acclimation responses are unlikely to dominate and indeed if occurring are would result in an underestimation of the differences between control and addition experiments. The experimental design of 24 hours, whilst suitable for investigating iron limitation, means that results are not truly representative of in situ photosynthetic parameters and should not be interpreted as such."

3) The sampling strategy is curious with one of the samples (station 5) collected under the mixed layer. In addition, at the end of the manuscript, one discover that there were three occupations of the transect with a total 6 stations sampled (?)

A consistent depth was chosen between 35-50m was chosen to minimise changes between experiments. The density profile of station 5 showed 3 distinct layers of water, suggesting that there were 2 mixed layers (which is known to occur in the Southern Ocean). The depth for the mixed layer presented in the text was the first depth at the criteria T – 0.2C was met. A secondary mixed layer was determined at 56m and the text has been reflected to indicate this, please see p. lines 252-254.

"The CTD density profile at experiment 5 was indicative of 2 mixed layers present, with the experiment performed above the deeper of the mixed layers (∼56m)."

4) There is no measurement of DFe in the samples themselves. What about contaminations of controls?

DFe was measured from the initial set up water as listed in table 1, but due to constraints with water volumes in the bottles DFe was not measured in the experimental bottles. However, contamination of control bottles would be evident as outliers in the data reported, and since all experimental results showed good exponential fits consistent with theoretical predictions of the response of production to varying light, we can safely assume there is very little to no contamination in any of the incubation bottles.

5) The measurements are not carried out in triplicate and it is therefore impossible to evaluate the precision in the estimation of the photosynthetic parameters.

This is correct and as such a statement has now been added to the methods to highlight the reader to this problem. Please see p. 4 lines 126-127.

"Due to physical constraints, the experiments were not conducted as triplicates, and as such evaluation of the precision/error within experiments is not possible."

6) Inconsistencies in the determination of the light attenuation coefficient (40% difference between the PAR profile and the empirical equation as a function of chlorophyll). It is then not known which estimate is used in the primary production model.

There are known caveats to using empirical calculations to derive Kd from Chl (Morel et al., 2007), and we highlighted this in the methods to draw attention to the readers. An additional line has been added to the discussion, p. 15 line 477.

"Biomass (Chl), as represented through Kd, did exert a large influence on PPwc (up to 59%, Fig. S4), but this influence could be overestimated due to potential errors in the calculation of Kd (Morel et al., 2007)."

As PAR profiles with depth were only available when CTDs were performed this limited the data set to only 6 profiles with latitude. Interpolation was performed using various methods to allow the calculation of PP at a higher resolution along the transect.

Also many problems in the expression of results:

1) a salinity change of 33.71 to 36.51 is considered "not distinct" (line 235). A chlorophyll range between 0.84 and 2.3 is considered as a "small range of variability", with individual values considered to be "low", indicating a total ignorance of the oceanography of this region.

I find the phrase 'total ignorance' highly offensive and a poor example of constructive criticism. Nonetheless, we thank the reviewer for drawing our attention to the salinity values, which were incorrectly reported from a malfunctioning CTD sensor. Bottle samples were instead used to provide the correct salinity range. Please see p. 9 line 244.

"Initial temperatures displayed a characteristic decrease from 10.80°C at the most northerly location to -1.51°C at the MIZ, whereas there were no distinct differences in salinity ranging from 33.70 to 33.88."

Chlorophyll concentrations in proximity to the islands and coastal regions can exceed 2.3 ug/L at the height of the growing season. Not displayed here but measured during the cruise were surface chlorophyll concentrations that exceed 5.0 ug/L up to a maximum of 11.5 ug/L. The text has been altered to reflect that this is a small range of variability in the chlorophyll concentrations relative to those measured during the transects. Please see p.14 lines 396-397.

"...when compared to the range of chlorophyll concentrations measured throughout the entire cruise (0.01 – 11.25 $\mu$g L-1)."

2) The presentation of the photosynthetic parameters (paragraph 3.2.) is surprising. It is written: "PE curves for carbon uptake (C) (Fig. 2, Fig. S1), summarised in Table 2, display consistent results with greater values of B and PB with the addition of iron compared to the control treatments (Fig. S2).", which is completely inconsistent with Figure S2. Moreover, the choice of the figure showing the effect of iron on the P-E

curve (figure 2) is at the limit of what is acceptable: it is the only good relation of this type on the 6 curves presented on figure S2. On top of that I do not understand how the parameters have been inferred from experimental data. I defy anybody to see significant changes in the photosynthetic parameters, related to iron enrichment, from the curves presented as supplementary material.

The parameters were determined following the standard equation of Platt et al. 1980 as outlined in section 2.2.

I can find no literature which suggests what is the acceptable limits for PE curves. All the resulting fits from the data presented here had r2 values >95% following multiple iteration non-linear least square fits.

The full results of the PE parameters are presented in table 2. Perhaps this is not made clear in Figure S2 due to the choice of colour scale, as such a new figure (S3) has been added to the supplementary material to highlight the differences between the treatments (which are small in the first three experiments in the SAZ but more substantial in the last three experiment south of the PF) as a bar chart – see below.

[Figure]

**Fig. 1.**

---

## Author Response (AR1)

SOCCO
Natural Resources and Environment
CSIR
Lower Hope Road
Cape Town
South Africa

Prof. Gerhard Herndl,
Associate Editor,
Biogeosciences

Wednesday, 28 June 2017

Dear Prof. Herndl,

**Response to reviewers for manuscript bg-2017-74**

On behalf of my co-authors and myself, we would like to thank both you and the reviewers for timely responses in commenting upon our manuscript entitled "Modelled estimates of spatial variability of iron stress in the Atlantic sector of the Southern Ocean". All comments were appreciated and have been taken into full consideration when making amendments to the revised manuscript submitted here.

In the document, we have highlighted changes (using the track changes function of MS Word) made following the reviewers' comments. We feel we have adequately addressed the reviewer's comments, and hope this manuscript is now acceptable for publication in Biogeosciences.

We outline our response to each of the reviewer's comments below.

Reviewer 1

The reviewer indicated that the results were definitely worthy of publication, addressing classical questions of iron limitation and primary production. Stating that although similar studies have been performed in the past, this study is unusual due to long latitudinal transect spanning the major Southern Ocean biogeochemical provinces.

1.  I think the results and conclusions obtained here need to be qualified as applying only to the initial responses of these communities to iron additions, and it should be explicitly recognized in the text that they cannot be applied to understand longer term community responses (which would also include taxonomic composition shifts).

    1.1. A statement to this effect has been added to the discussion to highlight that these results only reflect initial responses and do not take into account community shifts and longer term responses. Please see p. 13 lines 388-391.

2.  Likewise, the fact that in some of these short iron addition experiments PE parameters did not change (for instance, stations 1 and 2 in the SAZ, p. 10 lines 271-272 and Table 2) cannot be taken as evidence for lack of iron limitation at these stations. Much longer duration SAZ iron addition experiments published way back in 2001 show similar lack of changes in alphaB and PBmax, but in those same experiments the iron additions led to much higher biomass and to community composition changes- clear evidence that the community actually was iron-limited (Hutchins et al. 2001 JGR 106). Many other past studies have also conclusively confirmed ecological and physiological iron limitation of SAZ communities; one of the most highly cited is the SoFex in situ iron fertilization study of Coale et al. (2004, Science 304).

2.1. A statement has been added to discuss the potential for longer term relief from iron addition that may not have been achieved in this study. Please see p. 15 lines 449-451.

3.  …the gradient in Si availability this study covered was much greater than any gradient in N, P, or even Fe (Table 1, p. 5), and this is probably the reason that according to their pigment analyses haptophytes were dominant in the SAZ and diatoms south of the Polar Front (lines 245-247). Could changes in the makeup of these communities driven by Si availability have any influence on their PE results? This is probably worth considering briefly in the discussion.

3.1. A statement to address this has been added to the discussion, see p. 14 lines 406-412.

4.  Finally, as they briefly acknowledge on line 398, this cruise spanned two full months, and so in practice examined a seasonal shift as well as a latitudinal gradient. The relative importance of iron and light limitation (yes and Si limitation!) changes across the growing season in different ways in the various Southern Ocean regimes they examined (see a simplified diagram of this seasonal pattern in Fig 2 of Boyd et al 2010, L&O 55). It would be worthwhile to discuss this aspect of their study in more detail in the text.

4.1. The seasonal shifts in limitations have been added to discussion, please see p. 15 lines 461-468 and lines 477-480.

5.  Abstract lines 19-22. These quantitative values need to be better linked to the specific photosynthetic parameter each belongs to, it requires quite a bit of peering back and forth for the reader to figure out which number goes with which parameter. A simple rewording would be helpful.

5.1. Wording has been restructured to make it clearer, see p. 1 lines 18-22.

6.  Line 43. The Arrigo et al. 2013 reference on ice cover changes given here deals with the Arctic, not the Antarctic, and should be replaced. Please see p. 2 lines 47-48.

6.1. The Arrigo et al. 2013 reference has been removed and the following references have been inserted into this section.
6.1.1. Close, S. E. and Goosse, H.: Entrainment-driven modulation of Southern Ocean mixed layer properties and sea ice variability in CMIP5 models, Journal of Geophysical Research-Oceans, 118, 2811-2827, 10.1002/jgrc.20226, 2013.
6.1.2. de Lavergne, C., Palter, J. B., Galbraith, E. D., Bernardello, R., and Marinov, I.: Cessation of deep convection in the open Southern Ocean under anthropogenic climate change, Nature Climate Change, 4, 278-282, 10.1038/nclimate2132, 2014.
6.1.3. Zhang, J. L.: Increasing Antarctic sea ice under warming atmospheric and oceanic conditions, Journal of Climate, 20, 2515-2529, 10.1175/jcli4136.1, 2007.

7.  Line 123 and Figure 2. Obviously, the maximum irradiance of 400 used in the PE curves was still below photoinhibiting levels. It would have been interesting (if logistically challenging, as I admit!) to extend it out to higher irradiances to add some perspective on this end of the curve.

7.1. Indeed, the limitations of the incubator set up did prevent us from being able to determine potential levels of photoinhibition. Future experiments planned will encompass a change in experimental set up to try and achieve higher irradiances. Please see p. 10 lines 288-290.

8. Another thing I wish the authors had done was to get better taxonomic information than can be obtained by the simple class-level distinctions possible through HPLC pigment measurements along with CHEMTAX. Just because there were diatoms all along their transect doesn't mean they were ecologically or biogeochemically equivalent. In fact, SAZ diatoms tend to be small, delicate, lightly silicified pennates while diatoms south of the Polar Front are typically much more robust and silicified, and much more likely to be significant in export. It seems a shame to do all this work, and then be limited in the wider inferences that can be drawn due to having only bulk measurements of productivity and broad general classes of phytoplankton. Some more detailed taxonomic and functional information would have made the paper more useful and interesting.

   8.1. We agree with the reviewer that further taxonomic data would enhance certain aspects of this paper, as such a further study is being conducted using microscopy counts alongside coulter counter and HPLC data. However, this data analysis is ongoing and will not be available for this manuscript. The taxonomic data we do report however states that the SAZ is dominated by haptophytes not diatoms. In addition, we also present information on the dominant size structure from coulter counter effective diameter, which ranges in latitude from a minimum in the SAZ (4.29 um) to a maximum in the MIZ (8.59 um)– please see p. 9 lines 251 – 256. Diatoms only become dominant from experiment 4 onwards with subsequent changes in effective diameter.

9. Line 348. The Shi et al. 2007 reference is a study on the tropical $N_2$-fixing cyanobacterium Trichodesmium, and is not appropriate here. Please add a study on Southern Ocean phytoplankton, or at least on eukaryotic phytoplankton in general. Please see p. 11 line 366.

   9.1. Changed the reference to Raven 1990, Twining & Baines 2013, Quigg et al. 2003, Strezpek and Harrison 2004.

Reviewer 2

The reviewer felt like there were too many methodological problems associated with these results, as such we have tried to address each issue as follows.

1. No details are given on the pre-treatment of the incubation vials (ultra-clean conditions?)

   1.1. All experimental conditions were carried out in a class-100 clean container. Experimental bottles were pre-treated with detergent and acid (Hydrochloric) as per trace metal clean standards. Please see p.4 lines 1120 – 123.

2. The duration of the incubations (24 hours) does not make it possible to obtain an estimate of the in situ photosynthetic parameters of the natural phytoplankton because it is known that the adaptation time of these parameters in response to a change in light regime is on the order of the 2 to 6 hours. Within 24 hours, each incubated sample thus has ample time to adapt to the light intensity at which it is incubated. Nevertheless, these experimental values are used by the authors (apparently unaware of this major problem of different time scales between light acclimation and iron relief) in an extrapolation across the entire Atlantic area in order to evaluate the primary production of this sector.

   2.1. Statements to highlight this shortcoming have been added into the text, see p. 13 lines 388-398.

3. The sampling strategy is curious with one of the samples (station 5) collected under the mixed layer. In addition, at the end of the manuscript, one discovers that there were three occupations of the transect with a total 6 stations sampled (?)

3.1. A consistent depth was chosen between 35-50m was chosen to minimise changes between experiments. The density profile of station 5 showed 3 distinct layers of water, suggesting that there were 2 mixed layers (which is known to occur in the Southern Ocean). The depth for the mixed layer presented in the text was the first depth at the criteria $T - 0.2C$ was met. A secondary mixed layer was determined at 56m and the text has been reflected to indicate this, please see p 9. lines 259-261.

4. There is no measurement of DFe in the samples themselves. What about contaminations of controls?

   4.1. DFe was measured from the initial set up water as listed in table 1, but due to constraints with water volumes in the bottles DFe was not measured in the experimental bottles. However, contamination of control bottles would be evident as outliers in the data reported, and since all experimental results showed good exponential fits consistent with theoretical predictions of the response of production to varying light, we can safely assume there is very little to no contamination in any of the incubation bottles. Please see p.9-10 lines 274-279.

5. The measurements are not carried out in triplicate and it is therefore impossible to evaluate the precision in the estimation of the photosynthetic parameters.

   5.1. This is correct and as such a statement has now been added to the methods to highlight the reader to this problem. Please see p. 4 lines 131-132.

6. Inconsistencies in the determination of the light attenuation coefficient (40% difference between the PAR profile and the empirical equation as a function of chlorophyll). It is then not known which estimate is used in the primary production model.

   6.1. There are known caveats to using empirical calculations to derive Kd from Chl (Morel et al., 2007), and we highlighted this in the methods to draw attention to the readers. An additional line has been added to the discussion, p. 15 line 475-476.
   6.2. As PAR profiles with depth were only available when CTDs were performed this limited the data set to only 6 profiles with latitude. Interpolation was performed using various methods to allow the calculation of PP at a higher resolution along the transect – these interpolation methods are discussed in the methods, see p. 11 lines 326-334.

7. a salinity change of 33.71 to 36.51 is considered "not distinct" (line 235). A chlorophyll range between 0.84 and 2.3 is considered as a "small range of variability", with individual values considered to be "low", indicating a total ignorance of the oceanography of this region.

   7.1. An incorrect salinity value from the sensor was used in the text, bottle samples of salinity have now been used instead – see p. 9 line 244.
   7.2. Chlorophyll concentrations in proximity to the islands and coastal regions can exceed 2.3 ug/L at the height of the growing season. Not displayed here but measured during the cruise were surface chlorophyll concentrations that exceed 5.0 ug/L up to a maximum of 11.5 ug/L. The text has been altered to reflect that this is a small range of variability in the chlorophyll concentrations relative to those measured during the transects. Please see p.14 lines 415-416.

8. The presentation of the photosynthetic parameters (paragraph 3.2.) is surprising. It is written: "PE curves for carbon uptake (C) (Fig. 2, Fig. S1), summarised in Table 2, display consistent results with greater values of B and PB with the addition of iron compared to the control treatments (Fig. S2).", which is completely inconsistent with Figure S2. Moreover, the choice of the figure showing the effect of iron on the P-E curve (figure 2) is at the limit of what is acceptable: it is the only good relation of this type on the 6 curves presented on figure S2. On top of that I do not understand how the parameters have been inferred from experimental data. I defy anybody to see significant changes in the photosynthetic parameters, related to iron enrichment, from the curves presented as supplementary material.

8.1. The parameters were determined following the standard equation of Platt et al. 1980 as outlined in section 2.2. I can find no literature which suggests what is the acceptable limits for PE curves. All the resulting fits from the data presented here had r2 values >95% following multiple iteration non-linear least square fits.

8.2. The full results of the PE parameters are presented in table 2. Perhaps this is not made clear in Figure S2 due to the choice of colour scale, as such a new figure (S3) has been added to the supplementary material to highlight the differences between the treatments (which are small in the first three experiments in the SAZ but more substantial in the last three experiments south of the PF). Please see supplementary material.

We would like to thank you once again and hope that the changes made are sufficient to meet the requirements of publication in your journal.

Should you have any more comments or questions, then please do not hesitate to contact us.

Yours sincerely,

Thomas Ryan-Keogh
Postdoctoral Research Associate
SOCCO
Natural Resources and Environment
Direct tel: +27 21 658 2764
Email: Thomas.Ryan-Keogh@uct.ac.za

[revised manuscript text omitted]